# Dietary Responses of Dementia-Related Genes Encoding Metabolic Enzymes

**DOI:** 10.3390/nu15030644

**Published:** 2023-01-27

**Authors:** Laurence D Parnell, Rozana Magadmi, Sloane Zwanger, Barbara Shukitt-Hale, Chao-Qiang Lai, José M Ordovás

**Affiliations:** 1Nutrition and Genomics Laboratory, JM-USDA Human Nutrition Research Center on Aging at Tufts University, Agricultural Research Service, US Department of Agriculture, Boston, MA 02111, USA; 2Friedman School of Nutrition Science and Policy, Tufts University, Boston, MA 02111, USA; 3Skidmore College, Saratoga Springs, NY 12866, USA; 4Neuroscience and Aging Laboratory, JM-USDA Human Nutrition Research Center on Aging at Tufts University, Agricultural Research Service, US Department of Agriculture, Boston, MA 02111, USA; 5Nutrition and Genomics Laboratory, JM-USDA Human Nutrition Research Center on Aging at Tufts University, Boston, MA 02111, USA

**Keywords:** dementia, diet, flavonoids, gene–diet interaction, metabolic enzyme, taurine

## Abstract

The age-related loss of the cognitive function is a growing concern for global populations. Many factors that determine cognitive resilience or dementia also have metabolic functions. However, this duality is not universally appreciated when the action of that factor occurs in tissues external to the brain. Thus, we examined a set of genes involved in dementia, i.e., those related to vascular dementia, Alzheimer’s disease, Parkinson’s disease, and the human metabolism for activity in 12 metabolically active tissues. Mining the Genotype-Tissue Expression (GTEx) data showed that most of these metabolism–dementia (MD) genes (62 of 93, 67%) exhibit a higher median expression in any of the metabolically active tissues than in the brain. After identifying that several MD genes served as blood-based biomarkers of longevity in other studies, we examined the impact of the intake of food, nutrients, and other dietary factors on the expression of MD genes in whole blood in the Framingham Offspring Study (*n* = 2134). We observed positive correlations between flavonoids and *HMOX1*, taurine and *UQCRC1*, broccoli and *SLC10A2*, and myricetin and *SLC9A8* (*p* < 2.09 × 10^−4^). In contrast, dairy protein, palmitic acid, and pie were negatively correlated, respectively, with the expression of *IGF1R*, *CSF1R*, and *SLC9A8*, among others (*p* < 2.92 × 10^−4^). The results of this investigation underscore the potential contributions of metabolic enzyme activity in non-brain tissues to the risk of dementia. Specific epidemiological or intervention studies could be designed using specific foods and nutrients or even dietary patterns focused on these foods and nutrients that influence the expression of some MD genes to verify the findings presented here.

## 1. Introduction

Dementia is a multi-factorial syndrome that results in a progressive and accelerated decline in the cognitive functions of an individual, resulting in memory impairment and other symptoms that progress over time [1]. These other symptoms include the loss of memory, orientation, and communication, a diminished decision-making ability, and the gradual decline in the activities of daily life [2,3]. Dementia is central to vascular dementia, Parkinson’s, Alzheimer’s, and Huntington’s diseases [4,5]. As a chronic disease, dementia was the third-most attributable cause of mortality in the United States in 2017 [6] and by 2019, it ranked seventh globally [7]. This mortality burden is likely under-reported [8]. Additionally, dementia is a significant cause of disability [3]. Globally, 55 million individuals suffer from dementia [7] and since at least 1990 worldwide, and at least since 2000 in the USA, its incidence has been increasing [6,7].

Age is a major contributing risk factor for dementia as studies have shown that both the risk of and deaths attributable to dementia increase with an increasing age [6,9,10]. Other health and lifestyle risk factors include family history, ethnicity, social engagement, physical activity, both aerobic and strength training, sleep, stress, severe brain injury, dietary intake, the incidence of metabolic syndrome (MetS), and the diagnosis of cardiovascular disease [10,11,12,13,14,15], with the latter two being susceptible to diet. Importantly, these risk factors become more prevalent with increasing age, for example, as age increases, sleep is more disrupted [16], MetS increases [17], and physical performance decreases [18]. Short sleep duration is associated with a higher risk of MetS and MetS severity [19]. These conditions influence the rate of aging in the brain, which then dissociates the chronological age of the individual from the biological age of the affected organ. Complementary to epidemiological evidence are the numerous genetic association studies, either large-scale via arrays or family-based, that have identified a series of alleles of common genetic variants that are associated with the risk of cognitive decline and progressive neurological diseases such as Alzheimer’s and Parkinson’s disease, of which cognitive decline is a component. The genes identified by these approaches have been assembled into easily queried data resources [20,21].

Nutrition is a recognized primary influencer of metabolic processes as food is the source of the molecular components, coenzymes, cofactors, and prosthetic groups of the metabolism and metabolic enzymes [22]. For example, a recent review compiled evidence showing that a flavonoid supplementation, or a diet modified to include flavonoid-rich foods, may provide a therapeutic potential for aging individuals experiencing cognitive deficits resulting from neurodegeneration [23]. Yet, a sub-optimal diet and regular postprandial metabolic imbalances can contribute to metabolic dysfunction that acts as an insult to the proper function of the central nervous system [24]. Dietary patterns may be a more general but inclusive way to evaluate how diet affects cognition and its decline. For example, a Western diet, noted for elevated intakes of simple sugars, saturated fats, and processed foods, was shown to disrupt brain insulin signaling and promote the development of Alzheimer’s disease [25]. Additionally, a higher Dietary Inflammation Index, which is pro-inflammatory, was reported to be associated with a smaller total brain volume, a smaller total gray matter volume, and a larger lateral ventricular volume [26]. Alternatively, greater adherence to the MIND diet was observed in two US-based cohorts to be associated with a lower risk of dementia and a slower rate of cognitive decline [27], but this was not supported by an earlier study [28]. Similarly, an individual’s metabolic capacity and genetic constitution are important determinants of that individual’s nutritional requirements and health span [29]. Thus, it can be proposed that the diet exerts both positive and negative effects on neurological health and, by extension, on the rate of aged-related cognitive decline.

In some circumstances, a genetic variant has a severe impact on the metabolic capacity [30]. Many inborn errors of the metabolism (IEM) are disruptors of the metabolic activity in the brain and nerve tissues [30]. Phenylketonuria manifests as an abnormal phenylalanine metabolism resulting from deficient phenylalanine hydroxylase activity [31]. Quantitative in vivo 31P magnetic resonance spectroscopy elucidated changes in high-energy phosphorous metabolites accompanied by a diminished EEG background activity [31]. Glutaric acidemia type 1 disorder, caused by disruptive variations in *GCDH*, encoding the glutaryl-CoA dehydrogenase enzyme, blunts the energy metabolism in the mammalian brain [32]. Notably, the cascades of metabolic disturbances initiated by the respective genetic variants pertinent to these IEMs are often treated by specific, personalized adjustments to the patient’s diet [33,34]. In the GBA1 gene, the functional polymorphism E326K (rs2230288), which associates with an increased risk of Parkinson’s disease and Lewy body dementia, decreases glucocerebrosidase activity in vitro [35]. The ingestion of steryl-β-glucoside, a substrate of GBA1, found in Cycad seeds, which are part of the human diet in some cultures, was shown to enhance the in vitro aggregation and cytotoxicity of alpha-synuclein [36]. Besides diet [37,38], other lifestyle factors can exert a marked impact on cognition and the brain metabolism. These factors, including exercise [39,40], an enriched environment [39], educational attainment [8,10], and an active, socially engaged lifestyle [41,42], have been shown to reduce the risk of dementia, improve the cognitive function, and enhance specific metabolic functions in the brain. These observations indicate the responsiveness and plasticity of the various processes extending from different stimuli to brain health.

Several intensively studied dementia-affiliated genes such as *APOE* (apolipoprotein E), *GBA1* (glucosylceramidase beta 1), *LRRK2* (leucine rich repeat kinase 2), and *PINK1* (PTEN induced kinase 1) encode proteins that function in the human metabolism [43,44]. Several of these genes are annotated with an enzyme classification (EC) number, which ascribes a specific biochemical reaction to the protein. Thus, with an established link to metabolic functions, the purpose of this study was to perform an assessment of all such genes to identify which dementia-related genes have a documented function in the human metabolism, hereafter referred to as metabolism–dementia (MD) genes, and hence are more likely than others to be sensitive to or respond to diet, a modifiable lifestyle factor.

## 2. Materials and Methods

### 2.1. MD Genes

Genome-wide association study data [21], ClinVar [20], and the Online Mendelian Inheritance in Man (OMIM) [45] were queried with the term “dementia”, and then the genes were collected that were associated with different types of dementia, Alzheimer’s, and Parkinson’s diseases. Second, searches were performed of the Mammalian Metabolic Enzyme Database [43] and Enzyme at Expasy [44] to obtain genes coding for proteins that are associated with the human metabolism, plus rate-limited enzymes [46]. Lastly, all the collected genes and proteins were merged to yield a set of genes with an ascribed function in both the human metabolism and dementia in its various forms. Listed in Appendix A are the MD genes, with information on the gene’s symbol, identifier, and full name. Additionally, enzyme classification (EC) numbers [44] were allocated to a large number of the identified genes.

### 2.2. Gene Expression Data and Transcription Analysis

Data were mined from two resources: GTEx and the Framingham Heart Study.

**GTEx:** GTEx is a comprehensive public resource to study tissue-based gene expression and regulation derived from samples collected from 54 non-diseased tissue sites across more than 900 individuals. Data on all 93 MD genes were available in GTEx. For all MD genes, tissue-based gene expression data and expression quantitative trait loci (eQTL) data were mined from GTEx (Genotype-Tissue Expression) in July 2021 and September 2021, respectively (see Acknowledgements). Each MD gene served as a query whereby the expression data in the selected tissues were collected and tabulated for analysis. The selection of 12 metabolically active and diet-sensitive tissues entailed adipose (subcutaneous and visceral), colon (sigmoid and transverse), heart (atrial appendage and left ventricle), kidney, liver, skeletal muscle, small intestine, thyroid, and whole blood. Similarly, the expression data from 13 brain sections were collected. These sections were amygdala, the anterior cingulate cortex, caudate (basal ganglia), cerebellar hemisphere, cerebellum, cortex, frontal cortex, hippocampus, hypothalamus, nucleus accumbens (basal ganglia), putamen (basal ganglia), spinal cord (cervical c-1), and substantia nigra. Maximal median expression values were noted for each MD gene across both tissue sets.

**The Framingham Heart Study:** Detailed methods are provided elsewhere [47]. In brief, the participants of the Framingham Offspring Study (FOS, *n* = 2134), comprised of the children and spouses of the participants of the original Framingham Heart Study cohort, were interviewed and clinically examined approximately every 5 years. In this study, we used data from participants who took part in exam 8 (2005–2008). Only participants with completed diet and health assessment questionnaires were included in this study. These datasets were requested from dbGaP (https://dbgap.ncbi.nlm.nih.gov, with study accessions: phs000007.v28.p10 and phs000007.v25.p9; downloaded on 27 September 2017). The detailed methods for the analysis of the FOS gene expression data are provided elsewhere [48]. In brief, the fasted whole blood samples from the FOS participants at exam 8 provided the source material for the transcriptome analysis (from dbGaP under accession phe00002.v6). After quality control and normalization procedures [48], an accounting for the cell type heterogeneity across the samples was performed by calculating the principal components (PCs) based on the normalized expression of all the genes that passed quality control. The first 10 PCs were used in all the subsequent statistical analyses. In FOS, not all genes had data that passed quality control, or had data outright, and one of these is the important Parkinson’s and dementia gene *LRRK2*.

Data on the intakes of foods and nutrients, derived from the 126-item modified semi-quantitative food frequency questionnaire (FFQ), were processed according to Lai et al. [47]. Briefly, dietary exposures were classified as: (1) the daily absolute intake (amount) of nutrients, including macronutrients, fiber, vitamins, minerals, and bioactives. Macronutrients (i.e., fat, carbohydrate, and protein) were further expressed as percentages of the total energy intake; (2) individual food items (servings/day) as captured by the FFQ (i.e., 129 food items); and (3) food groups whereby individual food items were coalesced into 31 food groups. Alcohol intake (grams per day, or number of alcohol-drinking days per week), smoking (the number of cigarettes per day), and common medication use (prescribed for cardiovascular disease, hypertension, or type 2 diabetes) comprised other lifestyle exposures. Throughout this work, the sum total of these exposures is 291 dietary factors.

### 2.3. Correlation between MD Gene Expression and Dietary and Lifestyle Factors

Environment association analyses were conducted with three different groups of dietary intakes (Appendix A), measured in FOS exam 8, to determine the factors that correlate with MD gene expression. In a multiple linear model, the MD gene expression levels were set as the dependent variable and dietary exposure and lifestyle factors were set as the predictor variables. Covariates included the age at exam 8, sex, cell-type heterogeneity, and family relationship as a random effect. These analyses were developed and implemented in the SNP and VARIATION SUITE 8.9.0 (GoldenHelix Inc., Bozeman, MT, USA). To correct for multiple testing, a correlation matrix method was used by estimating the sum of the independent variables presented by the dietary intakes and lifestyle factors for each dependent variable, i.e., the MD gene expression [47]. The number of independent factors was 170. Thus, for each MD gene expression, we used Bonferroni adjustment, and the threshold for significance was *p* = 0.05/170, or 0.0003. The tests for the effect of age or sex on the MD gene expression employed a simple linear model, with the significance threshold set at 0.05.

### 2.4. Gene–Diet Interactions

The gene–diet interaction (GDI) tests for associations between the SNP genotype and MD mRNA level, as modified by specific dietary and lifestyle factors, was conducted as follows. The SNPs were mined from the GTEx expression QTL (eQTL) data for whole blood and was restricted to *cis*-eQTL and the expression of the MD genes. The eQTL data, including the SNP identifier, effect allele, and direction of the effect on the gene expression, were assembled for tests of gene–diet interactions. The genotypes of these SNPs were acquired in FOS data from dbGaP, as described above. The phenotypes of MD mRNA expression in blood were the same data as described above in “Gene expression data and transcription analysis.” The GDI tests were performed under a linear regression model with mixed effects, in the SNP and VARIATION SUITE 8.9.0 (GoldenHelix Inc., Bozeman, MT, USA). The covariates were age, sex, cell-type heterogeneity, and family relationship as a random effect. The significance threshold was set at 0.05 divided by the number of significant food intake correlations observed for the specific MD transcript (Appendix A).

## 3. Results

### 3.1. MD Genes as Aging Genes, as Rate-Limited Enzymes

#### 3.1.1. Aging Genes

Genes associated with dementia were collected from the databases of genetic associations and clinically relevant disease genes, as described in Methods Section 2.1, yielding a set of 93 metabolism–dementia (MD) genes (Appendix A). In addition, with the assignment of EC (enzyme classification) numbers to 76 of the MD proteins, the reactants and products of the MD-catalyzed reactions are generally known. Nine MD proteins have two or more EC numbers, indicating enzyme multi-functionality. As dementia is generally considered to be an age-related affliction, we assessed how many MD proteins show age-related effects of expression. We initially examined the MD gene expression in whole blood with the Genotype-Tissue Expression (GTEx) Project resource. GTEx is a repository of gene expression data across over 50 human tissues and supports the expression studies related to disease and genetic variation. We noted that 87 (94%) MD transcripts are detected in whole blood. Thus, we compared the MD gene set with the datasets from the GenAge [49] and InChianti [50] human studies in order to identify longevity-associated proteins in blood. This identified 23 MD proteins as longevity- or aging-associated (Appendix A). These include APOE (apolipoprotein E), F2 (thrombin), SOD1, SOD2 (superoxide dismutase 1 and 2), and PLAU (plasminogen activator urokinase). Seventy MD proteins were not classified in either dataset as longevity-associated blood biomarkers. It should be noted, however, that the GenAge and InChianti studies assessed only a small portion of the human proteome. Overall, these observations indicate that at least a subset of the MD proteins is considered as blood-based markers of aging.

#### 3.1.2. Rate-Limited Enzymes

Secondarily, a comparison with a set of 380 human rate-limited enzymes [46] indicated that 23 MD proteins have this attribute (Appendix A). The protein activity of a rate-limited enzyme could be acutely sensitive to slight changes in the reactant or product levels, i.e., the biochemical reaction constituents and perhaps those that are diet-derived [51]. There is a 1.6-fold enrichment of rate-limited enzymes in the MD dataset, which is statistically significant (Z-score = 2.50, *p* = 0.0062). Interestingly, 7 of the 23 MD proteins with the attribute of being rate-limited belong to the EC subclass 1.14.-.-, oxidoreductase. As there are 110 such oxidoreductases in the collection of 2448 human metabolic enzymes, this gives an enrichment of 6.7-fold, with a Z-score of 6.03 and a *p* value of 8.1E-10. The members of this subclass are oxidoreductases that act on paired donors, with an incorporation of or a reduction in the molecular oxygen that need not be derived from O_2_. The seven are CYP2D6, HMOX1, NOS1, NOS2, NOS3, PTGS2, and TH.

### 3.2. Gene Expression in Brain Sections and Selected Metabolically Active Tissues

Our primary interest was to investigate gene expression values in various tissues, reasoning that such information can indicate MD gene action distinct from what occurs in the brain. MD genes with substantial expression in tissues other than the brain might provide a contributory or supporting function in the development of the dementia phenotype affiliated with that gene. To investigate this, we obtained gene expression data for the MD genes from the GTEx resource. Data on the median values of the bulk expression for 13 brain sections and for a set of 12 key metabolically active tissues that are highly responsive to the diet were collected. Of 93 MD genes, 62 (66.7%) showed a higher median expression in any of the 12 metabolically active tissues, and 21 genes (22.6%) exhibited an over five-fold greater median expression in these tissues compared to the brain (Figure 1) (Appendix A). No brain expression data were reported for two genes: *MPO* (myeloperoxidase) and *SLC10A2* (sodium/bile acid cotransporter). Regarding the expression in the readily available tissue of the whole blood, a comparison of the expression in the brain to metabolically active tissues highlighted that 30 MD genes showed a median expression in the whole blood at a level minimally 0.5-fold of the maximum expression cataloged for any of the 13 brain sections or greater, up to 49-fold for *ALDH1A2* (aldehyde dehydrogenase 1 family member A2) (Appendix A). This value shows that a substantial number of MD genes are relatively highly expressed in the blood with respect to the brain tissue.

For several of the 12 metabolically active tissues, there are a number of genes whose median expression level is greater than the maximum median level reported for any of the 13 brain sections. This is particularly true for adipose, the small intestine, and thyroid, each having over 30 MD genes with this feature (Figure 2). Other than *MPO* and *SLC10A2*, a few genes relevant to dementia and encoding metabolic proteins are widely expressed at high levels across the panel of metabolically active tissues and at levels greater than in any brain section: *SOD2* (all 12 tissues), *ADH1C* (all except whole blood, alcohol dehydrogenase 1C), *ALDH1A2*, *CPT2* (carnitine palmitoyltransferase 2), *MAOA* (monoamine oxidase A), and *PLAU* (10 different tissues). At the other end of this tissue expression spectrum are 14 MD genes whose expression is greater in just one metabolically active tissue compared to all brain sections. These tissues and genes are subcutaneous adipose—*TBK1*; kidney—*ENO1*; liver—*AGT*, *APOC1*, *APOE*, and *SOD1*; skeletal muscle—*AMFR*, *NOS1*, and *PINK1*; small intestine—*ABCB1* and *GAPDHS*; thyroid—*MTHFR*, *TAF1*, and *WWOX*. Of these 14 genes, the median expression for 12 of them in the brain is between 0.525- and 0.997-fold of that in the other tissues. Incidentally, the expression of *CSF1R* (colony stimulating factor 1 receptor) in both adipose tissue types is greater than in all the brain sections and no other metabolically active tissue exhibits an expression greater than the brain. In contrast, although five different regions of the brain exhibit a robust expression of APOE, the values in liver are 1.9- to 2.8-fold higher. Similarly, the GBA1 expression in six of the 12 metabolically active tissues considered here was higher than the maximum mean value in the brain and, specifically, 2.4-fold greater in whole blood than the maximum of the 13 brain sections for which the data were available.

### 3.3. Correlations of MD Gene Expression in Blood with Dietary Intake, Age, and Sex

#### 3.3.1. Effects of Dietary Intake on MD Gene Expression

We hypothesized that metabolic enzymes encoded by genes with known connections to dementia and a high expression external to the brain could be responsive to signals deriving from the diet. Specifically, as the diet provides the molecular components of metabolic processes [22], and MD genes such as *APOE* are known to respond differentially according to diet [52], it follows that the levels of those components, as provided by the diet or the action of the gut microbiome, could influence the activity and effects across the set of MD enzymes. Thus, we determined the correlation of the MD gene expression in blood with over 170 independent dietary factors, plus a small set of other exposures, such as medication use for CVD, HT, and T2D and cigarette smoking. For these correlation tests, we examined the blood expression data from exam 8 of the FOS, with linear regression under a model adjusting for sex, age, family structure, the total energy intake, and the top ten principal components of the blood composition by cell type [47]. 

The transcript levels of 20 genes showed a total of 69 correlations that reached a statistical significance (*p* < 3.00 × 10^−4^) after correction for multiple testing (Appendix A). Of these 69 correlations, 47 had a positive beta (a higher expression with a higher intake of different food/nutrient items for 44 correlations; a higher expression with greater medicine or cigarette use for 3 correlations) and 22 had a negative beta (a lower expression with a higher intake of food/nutrient items for 13 correlations; a lower expression with greater medicine/cigarette use for 9 correlations). This difference in the distribution of beta direction by food vs. medicine/cigarette use was statistically significant (*p* = 0.0020, Fisher exact) (Figure 3) and shows the general effect of the food intake to increase the gene expression in the blood while medicine and cigarette use more often decrease the expression activity. Of the top responsive genes, we noted 23 distinct correlations that passed a statistical significance between the mRNA levels for heme oxygenase 1 (*HMOX1*) and the intakes of different specific food and nutrients and other exposures. Additionally, observed were 13 and 7 correlations, respectively, with the transcripts for *CSF1R*, and *SLC9A8*, encoding an exchange of extracellular sodium ions for intracellular protons. The top results are summarized in (Table 1). The correlations between the intake of specific food items and nutrients (i.e., dietary exposures) and the expression of well documented dementia genes *APOE* and *GBA1* did not reach a statistical significance.

To characterize the relationships between diet and MD gene expression, we determined the correlations between the intake values of specific food/nutrient items in the FFQ and expression. Corrections were made by adjusting the dietary data to 170 independent dietary factors [47]. We observed a positive relationship between the expression in the whole blood of *HMOX1* and the intake of tea (*p* = 7.73 × 10^−5^) and various flavan-3-ols, flavonols, and biflavonoids. Bananas, white wine, and manganese exhibited similar effects, while taking medication for type 2 diabetes correlated with a decreased expression (*p* = 1.09 × 10^−6^). The *SLC9A8* transcript levels corresponded positively with the intakes of myricetin (*p* = 1.46 × 10^−4^), apigenin (*p* = 7.28 × 10^−5^), and catechin (*p* = 4.00 × 10^−5^), a flavonol, a flavone, and a flavan-3-ol, respectively. The *SLC9A8* mRNA levels correlated positively with proanthocyanidin dimers (*p* = 8.10 × 10^−6^), but negatively with servings of pie (*p* = 2.80 × 10^−4^). *SLC10A2* encodes a bile acid transporter, notably for tauroursodeoxycholic acid (TUDCA) [53], and this transcript correlated positively with broccoli intake (*p* = 2.09 × 10^−4^). *UQCRC1* is a subunit of mitochondrial ubiquinol-cytochrome c reductase and we observed that its mRNA is positively associated with the intake of taurine (*p* = 1.47 × 10^−4^). Lastly, alcohol, a common effector of a change in gene expression, was noted to correlate positively with *MPO* and negatively with the prostaglandin-endoperoxide synthase *PTGS2* (*p* = 1.76 × 10^−4^; 3.53 × 10^−8^, respectively).

The expression values in the whole blood of *CSF1R*, *IGF1R,* and *PINK1* were also observed to correlate with the dietary factors (Appendix A). The *CSF1R* mRNA was negatively correlated with the intake of the saturated fatty acids palmitic acid and stearic acid, total fat, the use of T2D medication, and cigarette smoking (*p* < 1.20 × 10^−4^). Positive correlations were identified with lettuce (1.13 × 10^−4^), white wine (*p* = 2.35 × 10^−4^), and a number of vitamins and minerals, including folate, vitamin D, calcium, and manganese (*p* < 2.72 × 10^−4^). The expression of *IGF1R*, encoding the receptor of insulin-like growth factor 1, was observed as negatively correlated with the intake of dairy protein (*p* = 2.92 × 10^−4^). Similar to *CSF1R*, the expression of *PINK1*, a locus affiliated with the early onset of Parkinson’s disease encoding a serine/threonine protein kinase that localizes to mitochondria, was correlated with palmitic acid and stearic acid, total saturated fat, and the use of T2D medication, but all positively (*p* < 1.87 × 10^−4^). 

Of the various dietary factors and other exposures extracted from the FFQ that correlated with the expression of MD genes in the blood, some influence the expression of several genes. The three most negative beta values derive from type 2 diabetes medication, for transcripts *CSF1R*, *HMOX1,* and *NEK1* (*p* < 3.43 × 10^−5^). This influence extends to five of the eight most negative beta values, additionally for *GAK* and *TNFRSF21* mRNAs. Cigarette smoking was negatively correlated with three transcripts (Figure 3), *ALDH1A1*, *CSF1R,* and *PLCG2* (*p* < 5.98 × 10^−5^), but only with moderate effects relative to the dietary factors. Six transcript levels were noted to correlate significantly with alcohol intake. Of these, all were in a positive direction except for *PTGS2*. The only dietary factors affecting the levels of more than a single transcript were folate (*CSF1R* and *HMOX1*, both with positive beta), along with palmitic acid, stearic acid, and total saturated fat, as noted above. The taurine intake was observed to correlate with only one transcript, *UQCRC1* and positively, but its effect (the largest absolute beta value) is the strongest observed of all 69 correlations, regardless of the direction (Appendix A). 

#### 3.3.2. Age and Sex Effects on MD Gene Expression

Although age and sex can affect the gene expression and were controlled for in the analysis of the impact of the 291 dietary exposures on the MD gene expression, we examined the relationships with age and sex separately. Nine MD genes exhibited a significant age-associated differential expression in the blood (*p* < 0.05). Seven of these genes displayed an increased expression with age—*GAK*, *GBA1*, *INPP5D*, *INSR*, *PLAU*, *PRKCD*, and *UQCRC1* with *p*-values ranging from 1.44 × 10^−2^ for *UQCRC1* to 3.89 × 10^−4^ for *INPP5D*. Only a pair of genes—*CYLD* and *TRPM7*—exhibited a significant decreased expression with age, with *p*-values of 4.11 × 10^−4^ and 3.15 × 10^−3^, respectively. Twenty MD genes showed a significant sex-biased expression in blood. Of these, 11 exhibited a higher expression in males (*ALDH1A1*, *AMFR*, *CSF1R*, *FBP1*, *GSR*, *HMOX1*, *MPO*, *PLA2G4A*, *PRKAG2*, *SYNJ1*, and *TBK1*) with *p*-values ranging from 9.01 × 10^−3^ for *TBK1* to 2.79E-21 for *MPO*. Nine MD genes showed a higher expression in females (*ABCC4*, *CYLD*, *ENO2*, *EPHA1*, *MAOB*, *NEK1*, *PARP1*, *SOD1*, and *WWOX*) with *p*-values ranging from 1.09 × 10^−2^ for *NEK1* to 4.64 × 10^−6^ for *ENO2*.

### 3.4. eQTL Signals in Blood and the Allele-Specific Response to Dietary Factors

Our group maintains a long-running interest in nutrigenetics and personalized nutrition [52,54]. Recently, a new effort of the US National Institute of Health with its Nutrition for Precision Health initiative seeks to develop algorithms that predict individual responses to food and dietary patterns. Thus, we proposed to identify genetic variants that support an allele-specific response to the food or nutrient intake affecting the expression of the corresponding MD gene. This amounts to identifying gene–diet interactions (GDI) [54]. For those transcripts that exhibited significant correlations with a food item, nutrient, and other exposures, we mined the expression QTL (eQTL) SNPs for an expression in whole blood from GTEx data. Using those genotypes, when the data were available in FOS, we then assessed the interaction of the exposure as a modifier of the genotype–mRNA phenotype relationship. 

The only GDI that passed significance based on the number of dietary factors relating to the expression of the given transcript was rs17781378 on *GAK* mRNA as modified by T2D medication use (*p* = 0.043, beta = 0.0176, T = minor allele). Thus, no other tested GDIs between whole blood eQTL SNPs and the identified exposures were statistically significant. There are two main reasons for a lack of observable GDIs. There are no such interactions, or an ill-suited dataset was used to draw up a list of SNPs for the GDI tests. To the second point, the eQTL SNPs tested could drive basal expression and be less sensitive to transient signals from intermittent exposures (i.e., dietary and other). Essentially, the GTEx eQTL data are for basal or end-of-life transcription, while we are seeking to identify an allele-specific response to a specific dietary factor. Thus, it is not unexpected that a lack of identifiable and robust GDIs would be observed. It is plausible that such interactions exist but are weak. These two scenarios need not be mutually exclusive.

## 4. Discussion

In this work, we sought to describe the behavior of a select set of genes with documented involvement in dementia and encoding metabolic enzymes. Specifically, we present data showing that a majority (66.7%) of these metabolic–dementia genes are expressed to levels higher in metabolically active tissues external to the brain than in the brain itself. We then show that the expression in the blood of several MD genes is correlated with the intake of specific dietary factors, and those correlations often are in a direction that supports brain health or reduces a cognitive impairment or other disease conditions.

For these MD genes and proteins, our work provides further evidence that, between the brain and other tissues, a dynamic relationship of the expression and metabolic action of the encoded protein must be considered. An elevated MD expression in metabolically active tissues can compensate for the lower activity of the same gene and enzyme in the brain, thereby balancing the metabolic activity between the brain and other tissues. The control of this balance is likely to be less resilient with aging [55], upon insult from a sub-optimal diet, such as a high-fat diet [37] or one low in the intake of flavonoids [56], or as a consequence of genetic and epigenetic variants conferring the elevated risk of disease [57], for example, in a brain-specific promoter, for example, in a promoter or enhancer region active in one or more tissues of the brain, thereby regulating the expression of nearby genes. An imbalance in the MD gene and protein activity between MA tissue(s) and the brain could have consequences on neuronal health. Numerous studies have linked cerebrospinal, blood, and urine metabolites to cognitive decline, for example [12,58], with metabolite origins in the brain tissue as a result of cognitive decline, or in non-brain tissues that generally support a healthy brain metabolic capacity. Additionally, many dementia genes analyzed here were linked to dementia with genetic data. A genetic variant confers increased or, more often, reduced activity or could affect the expression in nerve or brain tissue [30]. Similarly, the allele-specific action of these genes and the encoded proteins in tissues external to the brain, then, could provide a kind of metabolic modulator to the suboptimal activity in the brain [33,34]. Hence, although enzyme kinetics must be considered, and likely for different tissues individually, there is a benefit in exploring the influence of externally derived exposures on the expression of the MD genes.

Many MD genes are expressed in tissues that experience the complexity of signals derived from the diet without the added filter of the blood–brain barrier. We hypothesized that diet likely influences to some degree the activity of the encoded MD proteins. A facet of this is the recent finding that liver fibrosis in middle-aged individuals was associated with an increased risk of incident dementia independent of shared risk factors, suggesting that liver fibrosis is not fully recognized as a dementia risk factor [59]. This finding connects the metabolic action in the liver to dementia.

For the MD gene *SLC10A2*, encoding a transporter of bile acids, including tauroursodeoxycholic acid (TUDCA), we observed a positive correlation between broccoli intake and an increased expression. In a mouse model of Huntington’s disease, the systemic application of TUDCA significantly reduced the striatal neuropathology of R6/2 transgenic mice. Specifically, the TUDCA-R6/2 mice exhibited reduced striatal atrophy, decreased striatal apoptosis, and improved locomotor and sensorimotor deficits, among other improvements [60]. The treatment of a mouse model of chronic Parkinson’s disease with TUDCA indicated it has neuroprotective properties, including protection against dopaminergic neuronal damage, preventing microglial and astrocyte activation, reducing dopamine and its metabolite 3,4-dihydroxyphenylacetic acid, and inhibiting alpha-synuclein aggregation [61]. Certain pathological hallmarks of Alzheimer’s disease, as modeled in amyloid precursor protein/presenilin1 transgenic mice, were attenuated by TUDCA [62]. Sulforaphane, present in broccoli, curtails neuroinflammation by activating KEAP1/NFE2L2 and inhibiting NFKB1 [63], but its effects on *SLC10A2* mRNA levels or protein activity are largely unknown. Our observation linking broccoli intake and *SLC10A2* expression largely agrees with these reports, where greater transporter activity is seen as beneficial.

Taurine, *UQCRC1*. Our analysis linked an increased taurine intake with the increased expression of UQCRC1, which encodes a subunit of the mitochondrial ubiquinol-cytochrome c reductase. The dysregulated expression of *UQCRC1* in the blood and skeletal muscle has been associated with Alzheimer’s disease [64,65]. Among different metabolites in a comparison with age- and sex-matched controls, Parkinson’s disease patients showed decreased cerebrospinal fluid levels of glutamate and taurine [58]. Taurine also has been shown to improve neuron injuries and cognitive impairment in a mouse Parkinson’s disease model by inhibiting microglial activation [66]. Neurochemical and behavioral evidence support a protective effect of caffeine and/or taurine in a rat model of Parkinson’s disease [67]. Lastly, a diet rich in taurine, cysteine, folate, vitamin B12, and betaine may lessen the risk of Alzheimer’s disease by boosting the brain synthesis of hydrogen sulfide [68]. Taken together, these observations show that taurine has certain beneficial effects on Alzheimer’s disease, Parkinson’s disease, and dementia, and our data links this amino acid to the mitochondrial enzyme UQCRC1. 

*HMOX1*. In this work, we noted that the increased intake of tea (not herbal) and various flavonoids, including flavanols and specifically quercetin and theaflavin, was correlated with an increased HMOX1 expression. Our observations agree with the published reports linking this gene, its transcriptional regulator NFE2L2 [69], and the flavanols with brain health and APOE activity. *ApoE4* transgenic mice exhibited lower hepatic Nfe2l2 nuclear protein levels and significantly lower mRNA and protein levels of Nfe2l2 target genes, including *Hmox1* as compared to *ApoE3* mice [69]. In *ApoE* knockout mice, quercetin significantly reduced various markers of inflammation and oxidation, and augmented the activity of vascular endothelial nitric oxide synthase, HMOX1 protein, and urinary nitrate excretion compared to the control. Theaflavin showed similar, although less extensive, significant effects [70]. The flavanol (-)-epicatechin has been demonstrated to provide cerebroprotection after a traumatic brain injury via NFE2L2-dependent and -independent pathways [71]. That the NFE2L2 regulatory elements in promoter regions respond to or are activated by various flavonoids, e.g., quercetin, kaempferol, fisetin, daidzein, luteolin, and apigenin [72], provides a link between our observation and this transcriptional regulator, its target gene *HMOX1,* and attributes of vascular dementia. Yet, while there is evidence that the benefit of an altered HMOX1 expression depends on an increase or decrease in different neuronal cell types [73], the benefits derived from other tissues are underexplored. Our results indicate that the flavanol-regulated expression of *HMOX1* in blood could be a mechanism that supports neuronal health, a finding that agrees with the assessment that a diet rich in flavonoids may confer therapeutic benefits to individuals suffering from cognitive deficits [23].

*SLC9A8*. Similar to *HMOX1*, we observed that the *SLC9A8* transcript levels corresponded positively with the intakes of certain flavonoids, specifically myricetin, apigenin, and catechin. A subgroup of mammalian sodium ion/proton exchangers (NHEs), of which SLC9A8 is one, consists of four isoforms with its members predominantly localizing to the membranes of the Golgi apparatus and endosomes. These organellar NHEs constitute a family of transporters that function in the regulation of luminal pH and in intracellular membrane trafficking as expressed, for example, in cell polarity development [74]. A mouse model of nonalcoholic fatty liver disease (NAFLD), in which the effects of a myricetin-supplemented diet were assessed, demonstrated regressed NAFLD effects in part via modulations in fecal butyric acid-related gut microbiota and the protection of gut barrier function [75]. *SLC9A8* is among four loci identified as contributing to the genotype–beta diversity of the gut microbiome [76], which invokes the gut–brain axis as a conveyor of dementia. Studies focused on the flavonoids showed that myricetin protected cells from endoplasmic reticulum (ER) stress and apigenin mitigated ER stress, respectively [77,78]. Together, these reports and our results support that certain flavonoids can mitigate dementia risk via an SLC9A8 function in membrane trafficking, perhaps in the gut.

*CSF1R*, *IGF1R,* and *PINK1*. The relationships between the observed correlations of diet and the *CSF1R*, *IGF1R,* and *PINK1* expression with regard to cognitive decline are less clear. CSF1R inhibitors offer a promising therapeutic strategy for neurodegenerative diseases by reducing microglial populations and the parallel neuroinflammation [79]. However, it must be noted that CSF1R inhibitors are typically supplied via the diet in preclinical research [80]. For example, food intake, as either high- or low-fat meals, altered the exposure as well as pharmacokinetics of CSF1R inhibition [80]. Thus, our observation that intakes of several vitamins, minerals, and leafy lettuce, all healthy dietary factors, correlate with an increased *CSF1R* expression are counter to the reports demonstrating the benefits of an inhibited expression, but those observations might be complicated by the food matrix. 

In our data, the intake of dairy protein correlated negatively with the blood expression of *IGF1R*, the receptor to insulin-like growth factor 1. Contrary to our finding, the down-regulation of *IGF1R* and its ligand IGF1 was observed in the hippocampus of rats with vascular dementia [81] and the treatment of mice with an IGF1R agonist alleviated the white matter damage and cognitive deficits after cerebral hypoperfusion [82]. However, our observation aligns with other reports. Neurons in Alzheimer’s disease that lack IGF1R exhibited a reduced accumulation of amyloid beta-containing autophagic vacuoles while the amyloid beta levels in plasma increased [83]. Two long-term cohorts from Japan suggest that the consumption of milk and dairy reduces the risk of dementia [84,85]. In a Chinese cohort, the highest consumption level of dairy was negatively associated with certain subclasses of mild cognitive impairment [86]. Lastly, IGF1R stimulating activity, as noted in 1014 participants in the Rotterdam study, was associated with dementia [87]. Together, the IGF1R data are complicated, but the nutritional epidemiological data are in good agreement with our observation and suggest the benefits of the intake of dairy may pass through IGF1R in tissues outside the brain. 

The regulation of *PINK1* transcription in response to stress is non-uniform, with activation or repression occurring based on different types of stressors, as reviewed [88]. However, the increased expression is generally viewed as conferring protection from mitophagy. We observe that the expression of *PINK1* correlates positively with intakes of lycopene, saturated fat, palmitic acid, and stearic acid. Although lycopene mitigated mitochondrial quality control disorder via regulating the SIRT1/PINK1/mitophagy axis and the mitochondrial unfolded protein response [89], the positive correlation of the intake of lycopene with the expression of *PINK1* was just beyond significance. A recent review acknowledged heterogeneity among studies, concluding that there is insufficient evidence and scarce data to state what the direct effects of lycopene are. Still, low circulating lycopene is a predictor of all-cause mortality, warranting further investigation into its relationship with cognitive longevity and dementia-related mortality [90]. 

We observed that intakes of several foods and nutrients generally correlated positively with the transcript levels of MD genes. In contrast, the use of medicines to treat hypertension and type 2 diabetes and smoking cigarettes exhibited the opposite correlations, often corresponding with decreased mRNA levels with increasing use. Whether this is an aspect particular to the metabolic–dementia genes investigated here, or the expression of metabolic enzymes in blood, or a more general contrasting behavior of food and nutrients compared to the stark effects of medicine and tobacco stimuli, remains to be examined in detail.

As with others, this study has its limitations. First, dietary factors can induce post-translational modifications (PTM) to proteins and those can affect the protein activity independent of changes in the gene expression. The datasets and cohort analyzed here do not provide PTM data, which introduces the potential caveat that gene expression is not a perfect proxy for protein activity. Second, although other lifestyle factors, such as exercise [39,40], educational attainment [8,10], and an active, socially engaged lifestyle [41,42], have been shown to reduce the risk of dementia, this study was not designed to assess the impact of these factors on MD gene activity. Third, myeloperoxidase, encoded by *MPO*, an MD gene, is expressed in the astrocytes of the Alzheimer’s disease brain where it supports the oxidation of phospholipids [91]. This observation underscores the gap between a common useful resource such as GTEx, with data on the gene expression in non-diseased tissues and the need for similar datasets from diseased tissues and individuals challenged with different diets. Fourth, in light of our observation that several MD genes showed high levels of expression in sigmoid colon tissue, this study is additionally limited by not assessing the impact of the microbiome, which is known to affect cognition [92]. Fourth, the results presented here should be investigated in other cohorts.

## 5. Conclusions

In conclusion, this report has shown that potential contributions to the risk of dementia exist by metabolic enzymes in non-brain tissues. The expression of genes with the dual aspects of coding for metabolic enzymes and relating to dementia can be influenced by the intake of specific foods and nutrients. These include specific flavonoids, taurine, broccoli, and tea. This information offers the opportunity to develop specific epidemiological or intervention studies.

## Figures and Tables

**Figure 1 nutrients-15-00644-f001:**
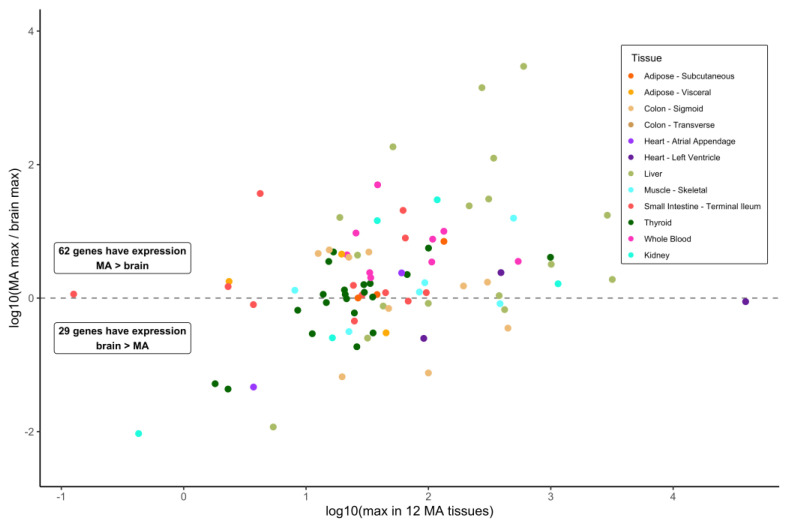
Expression of metabolism–dementia genes in 12 metabolically active tissues compared with 13 brain sections. Using data from the GTEx resource, metabolism–dementia (MD) gene expression is compared between 12 metabolically active (MA) tissues and 13 brain sections. For each MD transcript, the maximal expression value in MA tissues is plotted against a ratio of maximum expression in MA over maximum expression in brain. The MA tissue with maximal expression in all 12 tissues is indicated by dot color. For MD genes *MPO* and *SLC10A2*, no expression data in brain are available, and hence no ratio can be plotted.

**Figure 2 nutrients-15-00644-f002:**
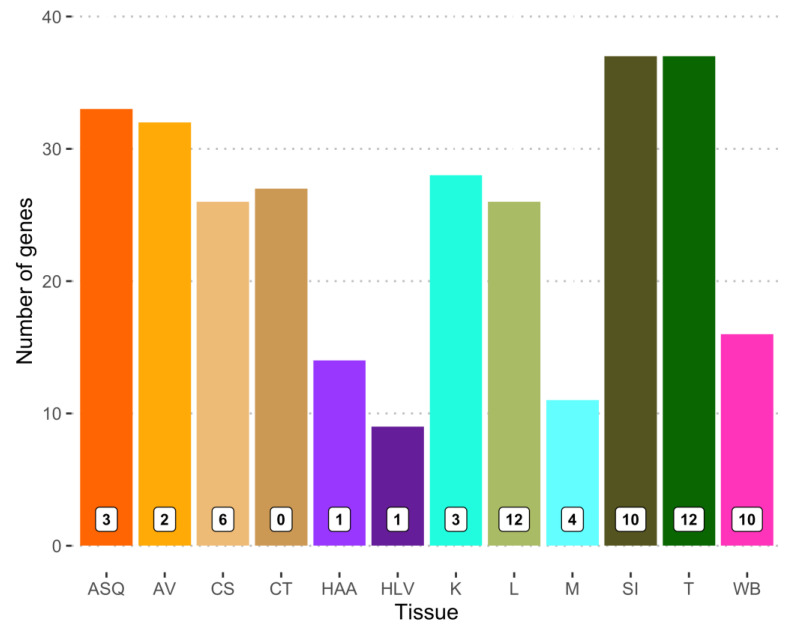
Comparison of MD gene expression across 12 metabolically active tissues with levels in brain. Bars indicate the number of MD genes whose median expression in that tissue is greater than the median expression in all 13 brain sections. Numerals in white squares are the number of MD genes with median expression highest of all 12 MA tissues and greater than all 13 brain sections. ASQ—adipose, subcutaneous; AV—adipose, visceral; CS—colon, sigmoid; CT—colon, transverse; HAA—heart atrial appendage; HLV—heart, left ventricle; K—kidney; L—liver; M—skeletal muscle; SI—small intestine; T—thyroid; WB—whole blood.

**Figure 3 nutrients-15-00644-f003:**
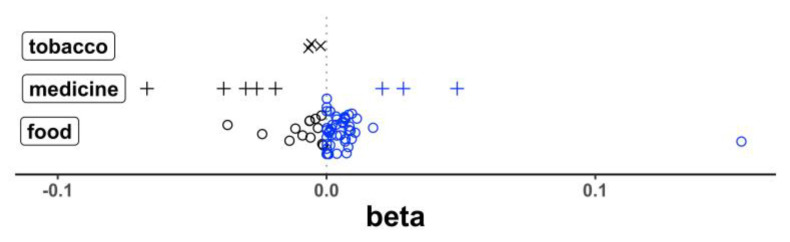
Distribution of correlations between dietary and other exposures and MD gene expression. Individual food items, nutrients, and other diet intakes that correlated significantly with expression of an MD gene more frequently showed a positive correlation (circles). In contrast, use of medicine to treat hypertension and type 2 diabetes (plus) or cigarette smoking (x) more often showed a negative relationship with gene expression, such that use results in lower expression levels. Blue color represents positive and black negative beta or correlation coefficient.

**Table 1 nutrients-15-00644-t001:** Top correlations between metabolism–dementia gene expression in blood and dietary factors.

Gene	Dietary Factor	*p*	Beta	SE *
*HMOX1*	Type 2 diabetes medication	1.09 × 10^−6^	−0.067	1.37 × 10^−2^
*HMOX1*	Manganese	1.47 × 10^−4^	0.017	4.55 × 10^−3^
*HMOX1*	Gallocatechin	7.83 × 10^−5^	0.0089	2.25 × 10^−3^
*HMOX1*	Alcohol	3.92 × 10^−5^	0.0081	1.96 × 10^−3^
*HMOX1*	Bananas	2.79 × 10^−4^	0.0069	1.91 × 10^−3^
*HMOX1*	Theaflavin, total	7.71 × 10^−5^	0.0068	1.71 × 10^−3^
*HMOX1*	Quercetin	5.57 × 10^−5^	0.0038	9.43 × 10^−4^
*HMOX1*	Tea	7.73 × 10^−5^	0.0036	9.15 × 10^−4^
*HMOX1*	Epicatechin 3-gallate	7.91 × 10^−5^	0.0018	4.61 × 10^−4^
*HMOX1*	Epigallocatechin	9.16 × 10^−5^	0.0013	3.41 × 10^−4^
*HMOX1*	Flavonoids, total	4.63 × 10^−5^	7.74 × 10^−5^	1.90 × 10^−5^
*HMOX1*	Folate, total	1.43 × 10^−4^	5.64 × 10^−5^	1.48 × 10^−5^
*IGF1R*	Dairy protein	2.92 × 10^−4^	−0.0016	4.40 × 10^−4^
*MPO*	Alcohol	1.76 × 10^−4^	0.0014	3.63 × 10^−4^
*PLCG2*	Cigarettes, number per day	5.98 × 10^−5^	−0.0022	5.47 × 10^−4^
*PTGS2*	Alcohol	5.82 × 10^−6^	−0.0089	1.96 × 10^−3^
*SLC10A2*	Broccoli	2.09 × 10^−4^	0.0082	2.21 × 10^−3^
*SLC9A8*	Pie, ready-made	2.80 × 10^−4^	−0.024	6.57 × 10^−3^
*SLC9A8*	Myricetin	1.46 × 10^−4^	0.0095	2.49 × 10^−3^
*SLC9A8*	Apigenin	7.28 × 10^−5^	0.0094	2.36 × 10^−3^
*SLC9A8*	Red wine	8.84 × 10^−6^	0.0035	7.82 × 10^−4^
*UQCRC1*	Taurine	1.47 × 10^−4^	0.154	4.06 × 10^−2^

* SE, standard error.

## Data Availability

The data presented in this study are available in Appendix A (all metabolism–dementia genes), with various attributes as described in Results available in Appendix A.

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
