# Peer review of "Dietary Responses of Dementia-Related Genes Encoding Metabolic Enzymes"

_nutrients, 2023, doi:10.3390/nu15030644_

Round 1
Reviewer 1 Report
Review of “Dietary responses of dementia-related genes encoding metabolic enzymes”
Major:
1. Section 3.1 seems to be methods rather than a result of the study, do not repeat the methods and clearly write the results. Furthermore, I don't believe this section of the results (3.1) requires its own subdivision. When addressing the other findings, you may mention that you prepared a table listing the genes connected with both dementia and human metabolism.
2. The protein activity of 23 MD genes as rate-limited enzymes was written in a generic and unclear manner. It wasn't enough to indicate that 23 MD genes had this feature without addressing the enzymes or the significance of that. If you intend to include this data in your study, explain why rate-limited enzymes are important and why it is important that some of the MD genes proteins are rate-limited enzymes. Add all necessary information or delete it.
3. Although you said in the introduction that "Several intensively studied dementia-affiliated genes such as APOE, GBA, LRRK2 and PINK1 encode proteins that function in human metabolism", I didn't observe any particular relevant data addressing these four genes in the results. My concern is, given that all of the genes listed in Table S1 are linked to human metabolism and dementia, why did you just highlight the mentioned four?
4. According to the study, the MPO and SLC10A2 genes were the only two that were not expressed in the brain. Since MPO is expressed in astrocytes in the human AD brain, I expected this information to be discussed in the discussion part. It would be interesting to discuss it.
Minor:
1. In line 42, “Dementia is central to vascular dementia, Parkinson, Alzheimer and Huntington diseases.”, please cite this sentence. Besides, they are Parkinson’s, Huntington’s, and Alzheimer’s diseases (please correct them in the whole manuscript). In lines 74 and 75, please add the needed references regarding Phenylketonuria disease. Please cite this sentence in line 89 “Several intensively studied dementia-affiliated genes such as APOE, GBA, LRRK2 and PINK1 encode proteins that function in human metabolism”.
2. When you first mention the genes (GBA, APOE…etc.), please provide their full names.
Author Response
On behalf of all co-authors, I wish to express my gratitude to the reviewers for their thoughtful and helpful comments. Those comments have stimulated noted improvements in the manuscript. IN addition, we wish to express thanks for patience on the parts of the reviewers and those at the publisher for understanding that the delays in completing revisions and responses were the result of administrative delays that were beyond the control of the authors.
Reviewer 1
Major:
1. Section 3.1 seems to be methods rather than a result of the study, do not repeat the methods and clearly write the results. Furthermore, I don't believe this section of the results (3.1) requires its own subdivision. When addressing the other findings, you may mention that you prepared a table listing the genes connected with both dementia and human metabolism.
Response:
Thank you - this is a good suggestion for streamlining the manuscript. We have shortened the part of the former section 3.1 vastly (by ~110 words) and incorporated the main point into the first true result of the work.
2. The protein activity of 23 MD genes as rate-limited enzymes was written in a generic and unclear manner. It wasn't enough to indicate that 23 MD genes had this feature without addressing the enzymes or the significance of that. If you intend to include this data in your study, explain why rate-limited enzymes are important and why it is important that some of the MD genes proteins are rate-limited enzymes. Add all necessary information or delete it.
Response:
Thank you for this observation and suggestion of improving this section. We have revised the text to include more information about this characteristic (lines 230-237): There is a 1.6-fold enrichment of rate-limited enzymes in the MD dataset, which is statistically significant (Z-score = 2.50, P = 0.0062). Interestingly, seven of the 23 MD proteins with the attribute of being rate-limited belong to EC subclass 1.14.-.-, oxidoreductase. As there are 110 such oxidoreductases in the collection of over 2440 human metabolic enzymes, this gives an enrichment of 6.7-fold, with a Z-score of 6.03 and a P value of 8.1E-10. Members of this subclass are oxidoreductases that act on paired donors, with incorporation or reduction of molecular oxygen which need not be derived from O2. The seven are CYP2D6, HMOX1, NOS1, NOS2, NOS3, PTGS2 and TH.
3. Although you said in the introduction that "Several intensively studied dementia-affiliated genes such as APOE, GBA, LRRK2 and PINK1 encode proteins that function in human metabolism", I didn't observe any particular relevant data addressing these four genes in the results. My concern is, given that all of the genes listed in Table S1 are linked to human metabolism and dementia, why did you just highlight the mentioned four?
Response:
We highlighted these four genes because there are numerous publications on the activity of these genes and their encoded proteins with regard to dementia, but they often are not considered as metabolic enzymes. In other words, the gene-dementia connection is far more commonly known than the gene-metabolism connection, but these are indeed metabolism genes. Hence, we simply had the intent to use something familiar to make clear the metabolism-dementia connection, and cannot possibly introduce all genes relevant to the topic and study itself. We do present data on PINK1 (lines 364-367), showing that its expression is correlated with intake of saturated fatty acids (P < 1.87E-04). Lastly, although not in the Results, we do state in the Methods that expression data for LRRK2 did not pass quality control in the FOS data (lines 160-162).
We have provided some new text in the Results that relates to these genes specifically (lines 282-287):
Although five different regions of the brain exhibit robust expression of APOE, values in liver are 1.9-to 2.8-fold higher. Similarly, GBA1 expression in six of the 12 metabolically active tissues considered here was higher than the maximum mean value in brain, and specifically 2.4-fold greater in whole blood than that maximum of the 13 brain sections for which data were available.
followed later by (lines 324-327):
Correlations between intake of specific food items and nutrients and expression of well documented dementia genes APOE and GBA1 did not reach statistical significance.
4. According to the study, the MPO and SLC10A2 genes were the only two that were not expressed in the brain. Since MPO is expressed in astrocytes in the human AD brain, I expected this information to be discussed in the discussion part. It would be interesting to discuss it.
Response:
First, it is important to note that the GTEx data include gene expression from RNA-Seq of non-diseased tissues. Thus, expression values in the human AD brain are not expected in that dataset. Second, that MPO is expressed in astrocytes in AD brain tissue in humans is a noteworthy observation that highlights a limitation of our work. Accordingly, we have supplemented the Discussion with information on this observation (lines 575-580): myeloperoxidase, encoded by MPO, an MD gene, is expressed in astrocytes of the Alzheimer disease brain where it supports oxidation of phospholipids [90]. This observation underscores the gap between a common useful resource like GTEx, with data on gene expression in non-diseased tissues, and the need for similar datasets from diseased tissues and individuals challenged with different diets.
Minor:
1. In line 42 {line 43}, “Dementia is central to vascular dementia, Parkinson, Alzheimer and Huntington diseases.”, please cite this sentence. Besides, they are Parkinson’s, Huntington’s, and Alzheimer’s diseases (please correct them in the whole manuscript). In lines 74 and 75 {lines 89-90}, please add the needed references regarding Phenylketonuria disease. Please cite this sentence in line 89 {line 108} “Several intensively studied dementia-affiliated genes such as APOE, GBA, LRRK2 and PINK1 encode proteins that function in human metabolism”.
Response:
We have added the requested reference (line 43): Dementia is central to vascular dementia, Parkinson, Alzheimer and Huntington diseases [4,5].
Within our group, we grappled with including the apostrophe S or not. This effort led us to the style and usage recommendations of the American Medical Association, specifically with regard to the use of eponyms, doi: 10.1093/jama/9780190246556.003.0015. We have chosen to follow that usage, and made note of this in the Acknowledgements.
Lines 89-90, formerly 74-75, cite reference 31.
References have been added (lines 108-110): Several intensively studied dementia-affiliated genes such as APOE (apolipoprotein E), GBA1 (glucosylceramidase beta 1), LRRK2 (leucine rich repeat kinase 2) and PINK1 (PTEN induced kinase 1) encode proteins that function function in human metabolism [43,44].
2. When you first mention the genes (GBA, APOE…etc.), please provide their full names.
Response:
We understand the reviewer’s point and acknowledge the usefulness of gene names.
Gene names have been added to several places throughout the manuscript, adding to those gene names that were included in the submitted version of the manuscript.
We are concerned that some paragraphs that mention several or more genes will become very long when all those full names are added. Thus, rather than inflate the manuscript’s word count, and to include the requested information for all genes, we have added the full names of all genes to Table S1, which now includes the HGNC gene symbol, Entrez gene identifier, and the full name of the gene. In order to guide readers, we added a statement to this effect in the Methods section that describes assembly of the MD gene set (lines 126-127): Listed in Table S1 are the MD genes, with information on gene symbol, identifier and full name.
In several other places throughout the manuscript, gene names are now given.
Reviewer 2 Report
Dear Authors,
Thank you for the opportunity to review your manuscript “Dietary responses of dementia-related genes encoding meta-bolic enzymes” the objective of this study was to perform an assessment of all dementia-affiliated genes to identify which dementia-related genes have documented function in human metabolism, hereafter metabolism-dementia (MD) genes.
The review of this paper is shown below.
Within the abstract you state: “Specific epidemiological or intervention studies could be designed using specific foods and nutrients that influence the expression of some MD genes to verify the findings presented here.” : It may also be appropriate to include a statement on Dietary patterns that may influence MD genes as particular nutrients ect in isolation have yet to be found to have a positive effect .
Line 40: a reference needs to be placed after time.
Within line 43: How do you then deal with the conversion of mild cognitive impairment to dementia?? Do you believe it is the same mechanism with the same gene expression?
Line 49: You state” Age is a major contributing risk factor for dementia as studies have shown that both the risk of and deaths attributable to dementia increase with increasing age” You may wish to state that the aging brain varies within individuals , and although age is a major cause it is not fundamentally the issue as it is related to the metabolic syndrome that you have not mentioned. Age is associated with a multifaceted set of circumstances and this needs to be highlighted within the text.
It may also be a consideration to discuss the age of onset of dementia as well as early onset dementia as a lead into your manuscript.
Line: 60-70 it would be very useful to give a brief overview of the impact of a western diet compared with what nutrients and what dietary patterns give major nutrients to the body and brain as the term “food” is very nonspecific and can relate to a western diet and not a nutritious diet.
Line 72: you state” In some circumstances a genetic variant has severe impact on metabolic capacity.” This may be correct as it needs a reference, also what are these genetic variants that related to Dementia?? The examples give do not and you may reconsider these specifically related to the topic being discussed.
Line 90: needs a reference after metabolism.
Line 108: “Gene expression data and transcription analysis”: It would be very helpful here if you place a table showing where you mined the data, what source, location and the number of samples as this is not clear. Also include the results of the FFQ.
Line 263: You state “ We hypothesized that metabolic enzymes encoded by genes with known connections to dementia and high expression external to the brain could be responsive to signals deriving from the diet.” Please explain this hypothesis further as there is no rationale for this within the text.
Line 282: What food intakes are you referring to?
Line 287 You state “Food, nutrient and other diet intakes……………..” please explain what is food??
Line 297: You state” specific food intake values” What are these and what do they relate to???
Line 325: You state” various dietary factors” what are these??
Line 339 ; You state “dietary factors” please explain in the context what are these factors.?
Line 388: “insult from a sub-optimal diet” please explain what this is in relation to the study outcomes.
Line 390 What is “brain-specific promoter.”?
Line 408: You state “This finding connects metabolic action in the liver to dementia.” This may have a correlation, and may be recognized as such, however, can may not be able to state causation, or it may be part of the overall metabolic impact? Appreciate your views.
Line 409: You state” Additionally, a meta-analysis of 13 studies with over 240,000 participants, indicated that intake of sugar-sweetened beverages was positively associated with the prevalence of cognitive disorders [45]. Together, these are but two examples of either diet or a metabolically active, non-brain tissue exerting an effect on the risk of dementia.” Please explain how this relates to your study as it does not seem relevant.
Line 444; What Tea?
Line : 491: What particular dairy protein are you referring to??
Line : 539: You state “dementia can be influenced by the intake of specific foods and nutrients.” Such as?
Author Response
On behalf of all co-authors, I wish to express my gratitude to the reviewers for their thoughtful and helpful comments. Those comments have stimulated noted improvements in the manuscript. In addition, we wish to express thanks for patience on the parts of the reviewers and those at the publisher for understanding that the delays in completing revisions and responses were the result of administrative delays that were beyond the control of the authors.
Reviewer 2
Dear Authors,
Thank you for the opportunity to review your manuscript “Dietary responses of dementia-related genes encoding metabolic enzymes” the objective of this study was to perform an assessment of all dementia-affiliated genes to identify which dementia-related genes have documented function in human metabolism, hereafter metabolism-dementia (MD) genes.
The review of this paper is shown below.
Within the abstract you state: “Specific epidemiological or intervention studies could be designed using specific foods and nutrients that influence the expression of some MD genes to verify the findings presented here.” : It may also be appropriate to include a statement on Dietary patterns that may influence MD genes as particular nutrients ect in isolation have yet to be found to have a positive effect .
Response:
We have revised this text to (lines 33-35): Specific epidemiological or intervention studies could be designed using specific foods and nutrients, or even dietary patterns focused on these foods and nutrients, that influence the expression of some MD genes to verify the findings presented here.
Line 40 {41}: a reference needs to be placed after time.
Response:
The requested reference has been added.
Within line 43: How do you then deal with the conversion of mild cognitive impairment to dementia?? Do you believe it is the same mechanism with the same gene expression?
Response:
MCI and dementia are multi-factorial conditions of neurological decline. Some mechanisms or entities (genes and their variants, proteins, metabolites, physiological and cellular attributes) are shared and others are distinct. Progression of the decline is also important such that altered expression or activity of an MD gene and its encoded proteins can be specific to one or a few of the stages of MCI to dementia conversion. Lastly, the clinical data we mined to obtain the 93 MD genes did not distinguish MCI from dementia, and thus it is very likely that genes with stronger or exclusive functions in MCI are included in this study.
Line 49 {line 50}: You state” Age is a major contributing risk factor for dementia as studies have shown that both the risk of and deaths attributable to dementia increase with increasing age” You may wish to state that the aging brain varies within individuals , and although age is a major cause it is not fundamentally the issue as it is related to the metabolic syndrome that you have not mentioned. Age is associated with a multifaceted set of circumstances and this needs to be highlighted within the text.
It may also be a consideration to discuss the age of onset of dementia as well as early onset dementia as a lead into your manuscript.
Response:
Thank you for recommending an expansion and clarification of the dementia-aging relationship. We have revised the text of this paragraph (lines 54-60), adding several new references:
Age is a major contributing risk factor for dementia as studies have shown that both the risk of and deaths attributable to dementia increase with increasing age [6,9,10]. Other health and lifestyle risk factors include family history, ethnicity, social engagement, physical activity, both aerobic and strength training, sleep, stress, severe brain injury, dietary intake, incidence of metabolic syndrome (MetS), and diagnosis of cardiovascular disease [10-15], which itself is susceptible to diet. Importantly, these risk factors become more prevalent with increasing age, for example, as age increases, sleep is more disrupted [16], MetS increases [17], and physical performance decreases [18]. Short sleep duration is associated with higher risk of MetS and MetS severity [19]. These conditions influence the rate of aging in the brain, which then disassociates chronological age of the individual from biological age of the affected organ. Complementary to epidemiological evidence are the numerous genetic association studies, either large-scale via arrays or family-based, that have identified series of alleles of common genetic variants that associate with risk of cognitive decline and progressive neurological diseases such as Alzheimer and Parkinson disease of which cognitive decline is a component. Genes identified by these approaches have been assembled into easily queried data resources [20,21].
Line: 60-70 {beginning on line 66} it would be very useful to give a brief overview of the impact of a western diet compared with what nutrients and what dietary patterns give major nutrients to the body and brain as the term “food” is very nonspecific and can relate to a western diet and not a nutritious diet.
Response:
After citing reference [24] in line 73, we revised the text to:
Dietary patterns may be a more general but inclusive way to evaluate how diet affects cognition and its decline. For example, a Western diet, noted for elevated intakes of simple sugars, saturated fats and processed foods, was shown to disrupt brain insulin signaling and promote the development of Alzheimer disease [25]. Additionally, a higher Dietary Inflammatory Index, which is pro-inflammatory, was reported to be associated with smaller total brain volume, smaller total gray matter volume, and larger lateral ventricular volume [26]. Alternatively, greater adherence to the MIND diet was observed in two US-based cohorts to associate with a lower risk of dementia and a slower rate of cognitive decline [27], but this was not supported by an earlier study [28].
Line 72 {87}: you state” In some circumstances a genetic variant has severe impact on metabolic capacity.” This may be correct as it needs a reference, also what are these genetic variants that related to Dementia?? The examples give do not and you may reconsider these specifically related to the topic being discussed.
Response:
The reference is now provided. We value the suggestion to turn the flow of the text back to dementia. We wished to provide some well-known examples of genetic variation causing a severe metabolic phenotype that manifests as diminished activity or function of any number of brain phenotypes. Further, in an attempt to strike a balance with the reviewer’s suggestion, we wish to retain the IEM examples and add text related to specific MD genes.
Revised text (lines 87-97): In some circumstances a genetic variant severely affects metabolic capacity. Many inborn errors of metabolism (IEM) are disruptors of metabolic activity in brain and nerve tissues [30]. Phenylketonuria manifests as abnormal phenylalanine metabolism resulting from deficient phenylalanine hydroxylase activity. Quantitative in vivo 31P magnetic resonance spectroscopy elucidated changes in high-energy phosphorous metabolites accompanied by diminished EEG background activity [31]. Glutaric acidemia type 1 disorder, caused by disruptive variations in GCDH, encoding glutaryl-CoA dehydrogenase, blunts energy metabolism in the mammalian brain [32]. Notably, the cascades of metabolic disturbances initiated by the respective genetic variants pertinent to these IEMs are often treated by specific, personalized adjustments to the patient’s diet [33,34].
We also added other text pertinent to diet and GBA1 (lines 97-101): In the GBA1 gene, the functional polymorphism E326K (rs2230288), which associates with increased risk of Parkinson disease and Lewy body dementia, decreases glucocerebrosidase activity in vitro [35]. Ingestion of steryl-β-glucoside, a substrate of GBA1, found in Cycad seeds, which are part of the human diet in some cultures, was shown to enhance in vitro aggregation and cytotoxicity of alpha-synuclein [36].
Line 90 {110}: needs a reference after metabolism.
Response:
References have been added.
Line 108 {129}: “Gene expression data and transcription analysis”: It would be very helpful here if you place a table showing where you mined the data, what source, location and the number of samples as this is not clear. Also include the results of the FFQ.
Response:
Each of the 93 MD genes was queried at GTEx for gene expression values and expression QTL (eQTL) SNPs (single or short nucleotide polymorphisms that associate with mRNA levels). At the GTEx resource, users of the genomic data are asked to place a specific text in the acknowledgement section, as we have done. We refer readers to the GTEx resource for specific information. We have added a brief statement as an overview of GTEx (lines 131-133): GTEx is a comprehensive public resource to study tissue-based gene expression and regulation derived from samples collected from 54 non-diseased tissue sites across more than 900 individuals. Data on all 93 MD genes were available in GTEx.)
In lieu of a table, we also have added some clarifications throughout this section of the Methods in order to guide readers to our approach in collecting data from this common resource.
Regarding the FFQ, first, we apologize for not stating in the Methods the size of the FOS cohort. This value of 2134 was stated in the Abstract, and is now also in the Methods (line 147). With over 500 items and 2134 participants, inclusion of the FFQ results here really is not feasible. A condensed version of the relevant FFQ information, sufficient to replicate our work, is provided in another very recent study from our group, reference 46 by Lai et al. The full data tables are available by application to dbGaP.
Line 263 {298}: You state “ We hypothesized that metabolic enzymes encoded by genes with known connections to dementia and high expression external to the brain could be responsive to signals deriving from the diet.” Please explain this hypothesis further as there is no rationale for this within the text.
Response:
We have included a rationale for this hypothesis, and added the following text (lines 300-304): We hypothesized that metabolic enzymes encoded by genes with known connections to dementia and high expression external to the brain could be responsive to signals deriving from the diet. Specifically, as the diet provides the molecular components of metabolic processes [22], and MD genes like APOE are known to respond differentially according to the diet [51], it follows that the levels of those components, as provided by the diet or the action of the gut microbiome, could influence the activity and effects across the set of MD enzymes.
What is referred to by food intakes?
Line 282: What food intakes are you referring to?
Line 287 You state “Food, nutrient and other diet intakes……………..” please explain what is food??
Line 297: You state” specific food intake values” What are these and what do they relate to???
Line 325: You state” various dietary factors” what are these??
Line 339: You state “dietary factors” please explain in the context what are these factors.?
Response:
We have revised the text on the description of these data in the Methods, The Framingham Heart Study (lines 163-173). We also refer the reviewer to the Methods section (Data on intakes of foods and nutrients), which describes the three categories of dietary exposures in the FOS dataset. The FFQ is a complex vehicle with data on nutrients (derived from analyses of the individual foods plus supplements (eg, vitamin and mineral pills), individual food items and food groups (individual food items are grouped). Second, we have modified the text in the lines indicated by the reviewer, and elsewhere, to clarify the ambiguity of our original statements. Essentially, we refer to the collection of the FFQ data as dietary (and other) exposures or factors, the latter term being used often when we observe an effect of that FFQ item on a given phenotype.
Line 388 {433}: “insult from a sub-optimal diet” please explain what this is in relation to the study outcomes.
Response:
We offer an explanation (from line 432) that includes a key connection to the Western diet pattern and the negative correlation with CSF1R, and to flavonoids, which relate to our findings for HMOX1 and SLC9A8:
, such as a high-fat diet [37] or one low in flavonoid intake [56],
Line 390 {435} What is “brain-specific promoter.”?
Response:
We have revised the text to (lines 435-437): for example, in a promoter or enhancer region active in one or more tissues of the brain, thereby regulating expression of nearby genes.
Line 408 {455}: You state “This finding connects metabolic action in the liver to dementia.” This may have a correlation, and may be recognized as such, however, can may not be able to state causation, or it may be part of the overall metabolic impact? Appreciate your views.
Response:
It is a difficult situation to defend the work of others. Still, we will offer some views. Foremost, we do not mean to imply that the reported correlation in the cited work represents causation, but the observation merits recognition and further investigation. The fibrotic liver will have an altered and compromised metabolic capacity. We also note that of the 93 MD genes, 34 genes exhibited expression in liver at a level minimally 0.5-fold of the maximum expression cataloged for any of 13 brain sections or greater, with 15 of those 34 genes at 2-fold or greater expression in liver. This indicates widespread expression in liver of these dementia-related metabolic enzymes. Our work draws the liver metabolism-dementia risk pair closer to one another, but, as stated, is not meant to imply causation. Rather, we simply show from another angle that many dementia-related metabolic enzymes exhibit robust expression in tissues external to the brain, which then warrants proposing specific experiments to rule in or out causation.
Line 409: You state” Additionally, a meta-analysis of 13 studies with over 240,000 participants, indicated that intake of sugar-sweetened beverages was positively associated with the prevalence of cognitive disorders [45]. Together, these are but two examples of either diet or a metabolically active, non-brain tissue exerting an effect on the risk of dementia.” Please explain how this relates to your study as it does not seem relevant.
Response:
Prompted by the reviewer’s comment on relevance to the study and upon second thought, we have removed this text and the affiliated references.
Line 444; What Tea?
Is this black, all or any tea?
Response:
This is 1 cup of tea, not herbal. Please see line 487.
Line: 491 {534}: What particular dairy protein are you referring to??
Response:
We are not referring to a specific protein in dairy foods, but the macronutrient, that is, cumulative protein found in dairy foods.
Line : 539: You state “dementia can be influenced by the intake of specific foods and nutrients.” Such as?
Response:
We have added the following after the statement in question (lines 588-590): These include specific flavonoids, taurine, broccoli, and tea.
Round 2
Reviewer 1 Report
I appreciate that the authors have revised the manuscript in light of the comments they've received.